# Expression of IMP3 and LIN28A RNA-Binding Proteins in Placentas of Patients with Pre-Eclampsia with and without Severe Features

**DOI:** 10.3390/biomedicines12040879

**Published:** 2024-04-16

**Authors:** Maja Barbaric, Katarina Vukojevic, Anita Kolobaric, Martina Orlovic Vlaho, Tanja Kresic, Violeta Soljic

**Affiliations:** 1Laboratory of Morphology, Department of Histology and Embryology, School of Medicine, University of Mostar, 88000 Mostar, Bosnia and Herzegovina; anita.kolobaric@mef.sum.ba (A.K.); violeta.soljic@mef.sum.ba (V.S.); 2Department of Anatomy, Histology and Embryology, School of Medicine, University of Split, 21000 Split, Croatia; katarina.vukojevic@mefst.hr; 3Department of Obstetrics, Gynecology University Clinical Hospital Mostar, 88000 Mostar, Bosnia and Herzegovina; martina.orlovic.vlaho@mef.sum.ba (M.O.V.); tanja.kresic@mef.sum.ba (T.K.); 4Faculty of Health Studies, University of Mostar, 88000 Mostar, Bosnia and Herzegovina

**Keywords:** pre-eclampsia, placenta, extravillous trophoblast, IMP3, LIN28A

## Abstract

Background: this study aimed to determine the expression of RNA-binding oncofetal proteins IMP3 and LIN28A in extravillous (EVT) and villous trophoblast (VT) cells of placentas from pre-eclamptic (PE) pregnancies to better understand the pathogenesis of PE. Methods: placental tissue of 10 patients with PE with severe features, 10 patients with PE without severe features and 20 age-matched healthy pregnancy controls were analyzed by immunohistochemistry, double immunofluorescence and qPCR. Results: We found a decreased percentage of IMP3-positive EVT cells in PE with and without severe features compared to that of the healthy control (*p* < 0.001). IMP3 expression was significantly low in VT of PE placentas compared to that of the healthy control (*p* = 0.002). There was no significant difference in LIN28A expression between groups of PE and the control group. Additionally, we noticed the trend toward downregulation of IMP3 mRNA and LIN28A mRNA in severe PE compared to that of healthy controls. Conclusions: We demonstrated that IMP3 expression is decreased in EVT and VT cells of placentas from pregnancies complicated with both PE with and without severe features. However, additional functional investigations are needed to clarify the role of IMP3 as a potential therapeutic target in the management of PE.

## 1. Introduction

The placenta serves a vital role as an intermediary between the mother and the developing fetus, facilitating the exchange of essential nutrients, gases and waste products. Therefore, its proper development and function are essential for fetal development and maintenance of pregnancy [1]. One of the key elements during early placentation is an invasion of the extravillous trophoblast (EVT) into the maternal tissue. From the 8th week until the mid-second trimester of gestation, cytotrophoblast (CTB) shell cells differentiate into the EVT that migrates from the villi tips and invades the uterus reaching the inner third of myometrium [2,3,4]. This process unique to human placentation mediates extensive morphological changes of maternal spiral arteries, resulting in the formation of high-capacitance, low-resistance vessels establishing uteroplacental blood circulation [5,6]. Insufficient or shallow EVT invasion may result in the impaired remodeling of spiral arteries and lead to the development of various pregnancy disorders such as preterm birth, pregnancy loss, intrauterine growth restriction (IUGR) and pre-eclampsia (PE) [7].

PE is a pregnancy-specific, progressive condition characterized by high blood pressure and signs of damage to other maternal organ systems. Most commonly, it occurs after the 20th week of gestation in women who previously had normal blood pressure values. It affects 2 to 8% of pregnancies worldwide, creating short- and long-term consequences for both mother and fetus. There are still large discrepancies in the classification of PE, but the one based on symptom severity into PE with and without severe features is commonly used [8,9]. Although PE is considered the most common cause of death in the fetal and perinatal period, its etiology has not been fully elucidated. Some previous studies suggest that PE develops through two stages, with the first one occurring during the early placentation and resulting in “maternal syndrome” and the second stage as the clinical manifestation of the disease after 20 weeks of gestation. With this regard, some researchers emphasize the importance of early EVT function disorders, including impaired proliferation, apoptosis, differentiation and invasion for the pathogenesis of PE [5,10,11].

The mechanism by which the EVT invades maternal uterine tissues is similar to the mechanism of tissue invasion in carcinogenesis. However, unlike tumors and metastasis, this process is strictly modulated by various factors including hormones, cytokines, membrane proteins, protein kinases and growth factors [12,13,14]. This similar behavioral pattern puts the “cancer-like” mechanisms into the focus of research on PE pathogenesis [5,15].

RNA-binding proteins IMP3 (insulin-like growth factor II mRNA-binding protein 3, IGF2BP3) and LIN28A are oncofetal-placental proteins, are highly expressed during embryogenesis and carcinogenesis and are found to play an important role in placental development and maintenance [16,17]. The high expression of IMP3 and LIN28A has been found in aggressive and undifferentiated tumors and is related to increased tissue invasion and metastasis development [18,19,20,21,22]. IMP3 is a post-translational regulator of trophoblast cell proliferation and differentiation and suggested to play an important role in regulating EVT invasion, which is considered as one of the critical moments in placental development [23]. Apart from the placenta, IMP3 is not expressed in healthy adult tissues. Only one work of research showed the decreased expression of IMP3 in PE pregnancies compared to that of gestational age-matched healthy controls, but it was conducted on EVT cell lines. It is suggested that low IMP3 expression leads to a shallow invasion of the EVT and the incomplete remodeling of spiral arteries [23]. LIN28A is a regulatory protein expressed in trophoblast cells, induces the differentiation of the cytotrophoblast into the syncytiotrophoblast and, similarly to IMP3, affects the invasive abilities of the EVT [24]. However, recent studies have presented conflicting data regarding the expression of LIN28A in placental tissue. While initial research indicated high expression levels of LIN28A in human term placentas, subsequent studies examining placentas from pregnancies affected with PE did not corroborate these findings, suggesting that LIN28A is also down-expressed in healthy term placentas [25,26]. Additionally, LIN28A may interact with IMP3, potentially leading to the stabilization of IMP3 during cell differentiation. Given the novelty of this finding, further research is needed to confirm the theory and better understand the mechanism of interaction between LIN28A and IMP3 [27].

Therefore, we wanted to explore the expression pattern of IMP3 and LIN28A in the human placenta in PE with and without severe features to investigate their significance in EVT and villous trophoblast (VT) cells compared to that of gestational age-matched healthy controls.

## 2. Materials and Methods

The material for analysis was placental tissue, including 10 placentas from women with PE with severe features, 10 placentas from women with PE without severe features and 20 healthy gestational age-matched placentas as a control group. Considering that the expression of the investigated proteins may vary depending on the gestational week, due to differences in gestational age among the groups of PE with and without severe features, samples from the control group were also divided into two groups [23]. Control 1 (*n* = 10) refers to the gestational age-matched control group for PE with severe features and Control 2 (*n* = 10) to the control group for PE without severe features (Table 1). Placentas were collected at the Department for Gynecology and Obstetrics at University Hospital Mostar. The tissue embedded in paraffin blocks was stored in the archives at the Laboratory of Morphology at School of Medicine University of Mostar where the research was conducted. The including criteria were singleton pregnancy and confirmed diagnosis of PE with or without severe features according to ACOG guidelines defined as an elevated blood pressure accompanied by proteinuria or signs of other organs’ damage. Elevated blood pressure refers to the systolic pressure ≥ 140 mmHg and diastolic pressure ≥ 90 mmHg measured twice at least 4 h apart after 20 weeks of gestation in a previously normotensive woman. The term proteinuria implies the occurrence of ≥300 mg proteins in 24 h urine or a protein/creatinine ratio ≥ 0.3. In the absence of proteinuria, PE is defined as gestational hypertension with damage to other organs, which implies the occurrence of one or more of the following parameters: thrombocytopenia, progressive renal failure, liver damage, pulmonary edema and newly developed cerebral and visual impairment. PE with severe features is defined as the newly established gestational hypertension in which the systolic pressure is ≥160 mmHg, the diastolic pressure is ≥110 mmHg measured twice in a time interval of 4 h with possible damage to the other organs mentioned above and PE without severe features as the systolic pressure < 160 mmHg and the diastolic pressure < 110 mmHg [8].

The following data regarding the mother and fetus in all three groups were observed: maternal age, gestational age, parity, body mass index (BMI) before the pregnancy, IUGR and systolic and diastolic blood pressure in current pregnancy, occurrence of cesarean section delivery, birth weight and puerperal complications (Table 1). Excluding criteria in all three groups were chronic hypertension, type 1 or 2 diabetes mellitus, chorioamnionitis, any inflammatory disease, multiple gestations and assisted reproduction methods.

### 2.1. Tissue Preparation

Placentas were collected after vaginal or cesarean section deliveries and fixed in 4% formalin. Tissue samples containing basal decidua (1 cm × 1 cm in size) were cut, washed in phosphate-buffered saline (PBS), dehydrated in alcohol, purified in xylol and then embedded in paraffin. Tissue sections were cut at 4µm on a rotatory microtome and mounted on silanized slides [28,29]. Placental tissue for all three groups was macroscopically normal.

### 2.2. Immunohistochemistry

Tissue sections were deparaffinized in xylol and rehydrated using descending concentrations of alcohol and the distilled water as we described previously [30]. Briefly, sections were incubated in 3% H_2_O_2_ for 15 min at room temperature to block endogenous peroxidase. Antigen retrieval was performed by heating the sections in EDTA buffer pH 9 in a microwave oven for 17 min. After cooling to room temperature and washing in PBS, primary antibody (LIN28A, IMP3) was applied and incubated at room temperature for 1 h after which it was washed three times with PBS. This was followed by the application of secondary antibody goat anti-rabbit/mouse (K5007, Dako, Glostrup, Denmark) and incubation at room temperature for 1 h. Sections were then washed three times with PBS, stained with diaminobenzidine substrate (DAB), washed twice in distilled water and contrasted with hematoxylin. After that, dehydration through the ascending concentrations of ethanol and twice in xylol was performed. Sections were mounted in mounting medium (Canada balsam) and covered with a coverslip. All tissue sections were examined using a ×40 objective on Olympus BX51 (Olympus, Tokyo, Japan) and photographed using DP71 camera (Olympus, Tokyo, Japan). Brown cytoplasmic staining was considered as the positive expression for IMP3 and LIN28A. Negative control tissue sections were subjected to the same procedure except they were incubated with PBS instead of primary antibody. The negative control exhibited only hematoxylin-stained nuclei. Lymph node tissue was used as a positive control. All sections were analyzed in a blinded manner by two observers (VS and MB).

### 2.3. Double Immunofluorescence Staining

Tissue sections were deparaffinized with xylol and rehydrated using descending concentrations of ethanol and distilled water. Antigen retrieval was performed by heating the sections in EDTA buffer pH 9 in a microwave oven for 17 min. After cooling to room temperature and washing in PBS, sections were incubated with a combination of primary antibodies (Table 2) for 1 h. Sections were then washed in PBS and incubated with a combination of secondary antibodies: goat anti-mouse rhodamine (AP124R; Jackson Immuno Research Lab, West Grove, PA, USA) diluted 1:300 in PBS and goat anti-rabbit FITC (AP132F; Jackson Immuno Research Lab, West Grove, PA, USA) diluted 1:300 in PBS for 1 h or anti-mouse IgG (H + L); F(ab’)2 Fragment Alexa Fluor^®^ 488 Conjugate (4408S; Cell Signaling Technology Inc, Danvers, MA, USA) diluted 1:500 in PBS and anti-rabbit IgG (H + L); F(ab’)2 Fragment Alexa Fluor^®^ 594 Conjugate (8889as; Cell Signaling Technology Inc, Danvers, MA, USA) diluted 1:500 in PBS for 1 h. After rinsing in PBS, sections were counterstained with DAPI and coversliped (Immuno-mount; ANNEX C/11.2017 6 Shandon Inc., Pittsburgh, PA, USA). Negative control tissue sections were subjected to the same procedure except they were incubated with PBS instead of primary antibodies. The negative control exhibited only DAPI-stained blue nuclei. Lymph node tissue was used as a positive control. All sections were analyzed in a blinded manner by two observers (VS and MB). Sections were examined using the fluorescence microscope Olympus BX51 (Olympus, Tokyo, Japan) and photographed with DP71 camera (Olympus, Tokyo, Japan). IMP3 and LIN28A percentage was presented as a proportion of positive cells in the total cell count of EVT, whereas the expression in EVT and VT was presented semi-quantitatively (+1, ++2 or +++3). Double immunofluorescence staining with CK7 was used to distinguish decidual from EVT cells. Double immunofluorescence staining with hCG was used to distinguish CTB from SCTB cells. Both cell types of VT were positive for CK7.

### 2.4. qPCR

RNA isolation from FFPE tissue was performed using Sigma Aldrich Gen Elute TM FFPE RNA Purification kit. The total RNA concentration was measured by Qubit 4 Fluorometer HS RNA kit. qPCR analysis was performed on Applied Biosystems 7500 Real-Time PCR instrument (Applied BiosystemsFast 7500, Waltham, MA, USA) using Taqman^®^ Universal Master Mix II (Applied BiosystemsFast 7500, Waltham, MA, USA) containing AmpEraseuracil-N-glycosylase and the passive reference dye ROX. Primers and probes Taqman^®^ gene expression assays for human *IMP3, LIN28A*, 18sRNA (housekeeping gene) and glyceraldehyde-3-phosphate dehydrogenase (GAPDH; housekeeping gene) were supplied by Applied Biosystems (Applied BiosystemsFast 7500, Waltham, MA, USA; Hs00251000_s1, Hs04189307_g1, Hs03003631_g1 and Hs02758991_g1, respectively). Taqman real-time PCR was performed using approximately 50 ng cDNA template, 1 μL Taqman^®^ gene expression assay (Applied BiosystemsFast 7500, Waltham, MA, USA) and 10 μL Taqman^®^ universal master mix (Applied BiosystemsFast 7500, Waltham, MA, USA) in a final volume of 20 μL. The PCR protocol used involved heating for 2 min at 50 °C for uracil-N-glycosylase activation followed by heating for 10 min at 95 °C for polymerase activation and 40 cycles of amplification (15 s at 95 °C and 1 min at 60 °C). We performed duplicate PCRs per gene and per cDNA sample. A nuclease-free water instead of adding cDNA template served as a negative control and was included in each experiment. The 2-ΔΔCT method was used as the method of relative quantification.

### 2.5. Statistical Analysis

Statistical analysis was performed using GraphPad Prism 8.00 for Windows (GraphPad software, San Diego, CA, USA). Data were presented as median ± IqR. Kruskal–Wallis and Dunn’s post hoc tests were used for statistical analysis. *p* < 0.05 was considered as significant.

The total number of participants was calculated based on statistical significance (*p* = 0.001) and study power of 0.90 using the http://www.stat.ubc.ca/~rollin/stats/ssize/ online program. Based on input parameters (arithmetic mean and standard deviation) sample sizes of 4 and 5 placentas per group were required.

## 3. Results

### 3.1. Quantification IMP3 and LIN28A Expression in the EVT and VT of Placentas with PE with and without Severe Features Compared to That of Normal Healthy Pregnancies

Results obtained from immunohistochemistry analysis indicated a decreased percentage of IMP3 in the EVT of placentas in PE with and without severe features compared to that of the control (Figure 1 and Figure 2).

Results obtained by double immunofluorescence staining analysis indicated significantly decreased IMP3 expression in both EVT (Figure 3) and VT cells (Figure 4) in PE with and without severe features compared to that of healthy pregnancy controls, while we did not find significant difference in the expression of LIN28A in PE with and without severe features compared to that of the healthy control group (Figure 5 and Figure 6) (data shown in Table 3).

As we did not find a significant difference in the percentages of LIN28A (figure not shown) and IMP3-positive cells (Figure 1) between PE with and without severe features, for the further statistical analysis of staining intensity, we observed groups of PE with and without severe features as a single group (Table 3).

### 3.2. qPCR Analysis of IMP3 and LIN28A in Whole Placental Tissue

Performing the qPCR analysis, we noted the trend toward downregulation of *IMP3* mRNA and *LIN28A* mRNA in PE with severe features compared to that of the healthy controls, but the difference was not statistically significant (Figure 7).

## 4. Discussion

Our study aimed to determine the expression of oncofetal proteins IMP3 and LIN28A in EVT and VT cells of placentas from PE pregnancies to better understand the etiology of PE. As IMP3 and LIN28A play important roles in the proper development of placenta and remodeling of spiral arteries, it is reasonable to expect that their altered expression may lead to the development of pregnancy complications, including PE. However, studies on this topic are limited. Although PE is a severe complication known for hundreds of years, numerous studies are still struggling to provide better insights into the background of the disorder. Histology including immunohistochemistry and immunofluorescence is suggested to be an important tool of research in this field [31].

The results obtained regarding the expression of IMP3 in EVT and VT cells are in line with the previous findings that demonstrated the decreased expression of IMP3 in PE. Namely, our research on EVT cells of human term placentas confirmed those that Li et al. obtained using the cell lines [23,32]. Given that we found the down-expression of IMP3 comparing both the PE groups (with and without severe features) with the control group but not between the groups of PE with and without severe features, we can propose that the severity of PE does not depend on IMP3 expression, but the downregulation of IMP3 could serve as an important indicator for the development of the disease. With more detailed subclassification of the healthy control group based on the week of gestation, we noticed a trend of decreased IMP3 expression in healthy placentas from later gestational age compared to that of placentas from the earlier stage of gestation. Regardless, the IMP3 expression is still significantly higher in healthy placentas compared to that of gestational age-matched placentas from both PE with and without severe features, similarly to what was proposed by Li et al. [23].

Initial investigations suggested that LIN28A is highly expressed in placental tissue during term delivery [25]. Subsequent studies, examining various pregnancy complications have supported this hypothesis and demonstrated significantly decreased LIN28A expression in EVT cells from pregnancies complicated with PE and IUGR [33,34]. However, our study found no significant difference in LIN28A expression between the forms of PE with and without severe features compared to that of the control group, although we noticed a trend of LIN28A downregulation. This finding is consistent with the previous research conducted by Canfield et al. Similarly, we noted relatively low expression of LIN28A even in the healthy control group [26]. Such large discrepancies may be the result of lower expression levels in placental tissue during the later stages of gestation compared to those of the placental samples from the first trimester. Moreover, the results that differ from those of our study stem from research conducted on different cell lines rather than on human placental tissue, as was the focus of our investigation.

Our research on IMP3 and LIN28A was mainly focused on placenta-derived factors contributing to PE, as they present the greatest challenge in understanding the pathogenesis of the disease, especially considering that most cases remain idiopathic. However, it is widely accepted that PE is a heterogeneous condition, which is partly maternally derived [11]. Therefore, we also investigated maternal risk factors that could facilitate the early recognition of the disease. Some previous studies already recognized a history of PE, chronic hypertension, diabetes mellitus, multiple gestations and assisted reproduction methods as the prominent risk factors that can trigger the development of PE. Intending to further investigate risk factors, these were excluded from our study. Similarly to previous findings, we did not find a correlation between maternal age or pregnancy losses and the onset of PE [35,36,37]. Some authors indicated the importance of gestational BMI and a number of previous pregnancies for the development of PE, but our study did not reveal the same results [38]. The discrepancy observed may stem from the different times when BMI index was calculated, as we took only the value noted at the beginning of the pregnancy. Our results revealed different courses and outcomes of pregnancy not just between the control group and PE but also in the form of the disease without severe features compared to PE with severe features. Specifically, incidences of IUGR and postpartum complications were more common in cases of PE with severe features. While some authors have previously contended that IUGR is only consistently more prevalent complication in cases of PE with severe features, additional studies have revealed that the pregnancy outcomes associated with PE with severe features vary across different populations. These differences are influenced by factors such as the lifestyle and health status of the women within those populations [39,40]. Like in our previous research, here, we demonstrated that all pregnancies complicated by PE with severe features result in cesarean section delivery. While it has been previously established that cesarean section does not confer additional benefits over vaginal delivery in cases of PE with severe features, some authors still suggest that the occurrence of cesarean section delivery is high in PE with severe features and PE with severe features accompanied by complications. Therefore, this outcome may be partly influenced by the practices within institutions [41,42].

The limitation of our study was the small sample size. We concur that further studies should encompass a larger sample size and functional investigations to facilitate more precise conclusions.

The absence of identifiable clinical predictors and maternal risk factors emphasizes the importance of investigating intrinsic factors and human placental tissue. Although it is a hardly accessible tissue, future research should be focused on investigating IMP3 expression on EVT cells in first-trimester human placentas. Some studies previously suggested the role of LIN28A in stabilizing the expression of IMP3 [27]. Given that we found decreased LIN28A expression within both healthy and PE placentas, it is necessary to seek out a novel candidate partner that can affect the reduced invasion of EVT and consequently contribute to the development of PE.

## 5. Conclusions

The expression of IMP3 but not LIN28A was significantly lower in VT and EVT cells of placentas from pregnancies complicated by PE with and without severe features compared to that of the healthy control. The dysregulation of IMP3 expression may disrupt placental development and function leading to impaired fetal growth and maternal health issues characteristic of PE. Further research into this protein may provide insights into the pathophysiology of PE and identify potential biomarkers or therapeutic targets.

## Figures and Tables

**Figure 1 biomedicines-12-00879-f001:**
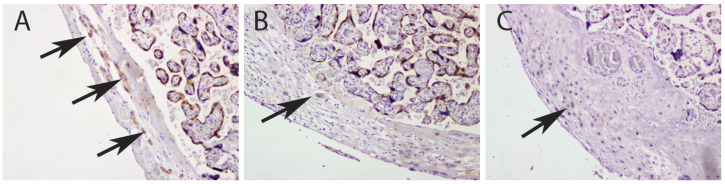
IMP3 expression in EVT cells (arrows) of decidua basalis in the placenta of a healthy control (**A**), PE without severe features (**B**) and PE with severe features (**C**).

**Figure 2 biomedicines-12-00879-f002:**
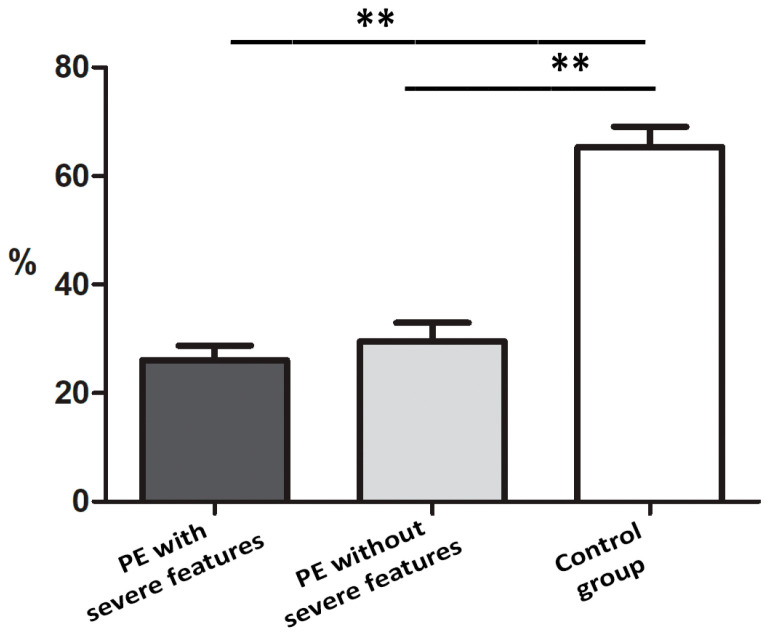
Percentage of IMP3-positive EVT cells of placentas from pregnancies complicated with PE compared to that of the control group. The data were presented as mean ± SD. Mann–Whitney test was used ** *p* < 0.001.

**Figure 3 biomedicines-12-00879-f003:**
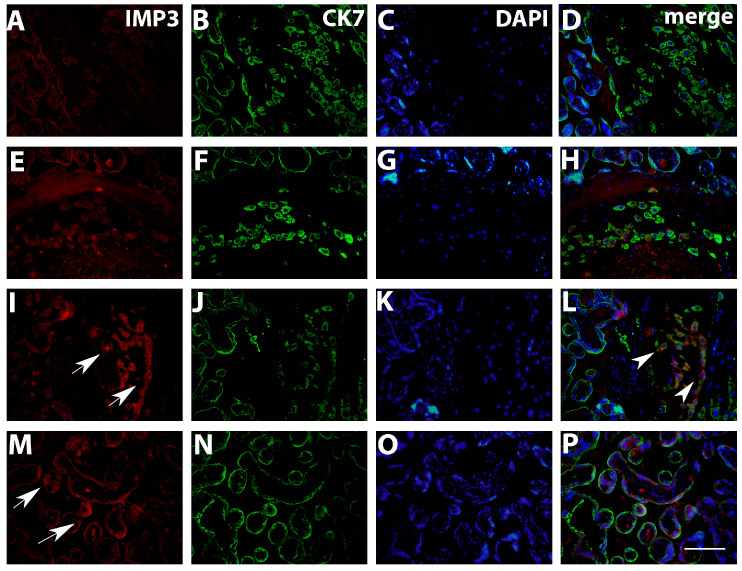
Expression of IMP3 in EVT (**A**–**H**) in PE with severe features, EVT (**I**–**L**) and VT (**M**–**P**) in healthy control group assessed by double immunofluorescence staining. Red cytoplasmic staining shows IMP3 expression (arrows), while green cytoplasmic staining represents VT and EVT cells positive for CK7. After merging with nuclear DAPI staining, significantly higher number of IMP3-positive cells in EVT is seen in the control group (arrowhead). Magnification × 40. Scale bar = 25 µm.

**Figure 4 biomedicines-12-00879-f004:**
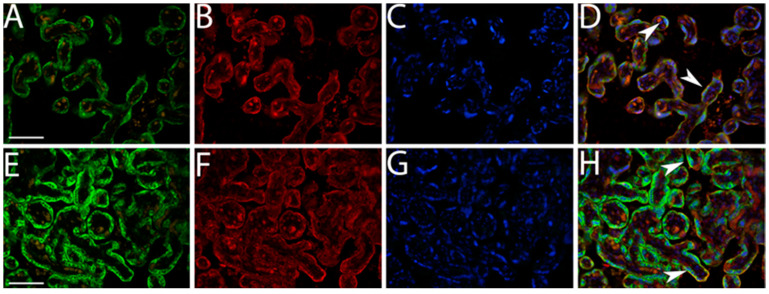
Expression of IMP3 in SCTB in PE with severe features (**A**–**D**) and healthy control group (**E**–**H**) assessed by double immunofluorescence staining. Green cytoplasmic staining shows IMP3 expression, while red cytoplasmic staining shows SCTB cells of chorionic villi positive for hCG. Merging with nuclear DAPI staining revealed significantly decreased staining intensity between PE and the control group (arrowhead). Magnification × 40. Scale bar = 25 µm.

**Figure 5 biomedicines-12-00879-f005:**
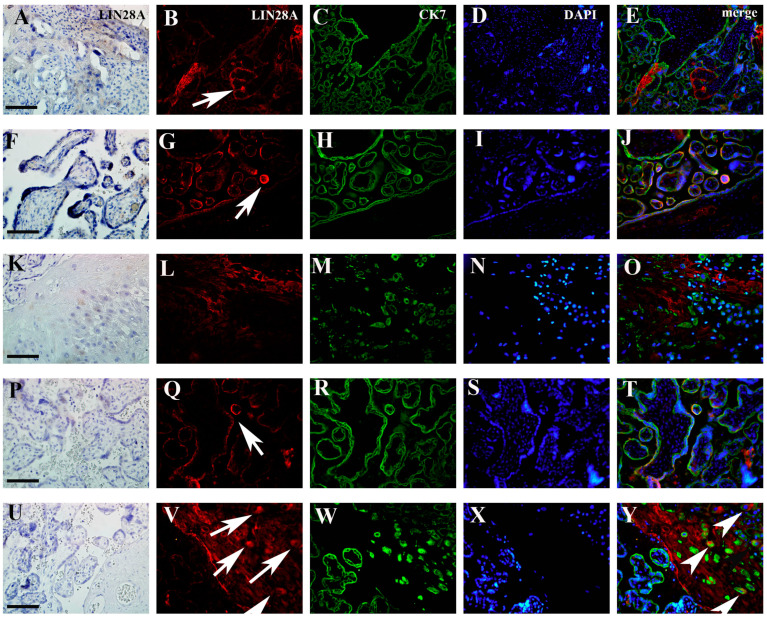
Expression of LIN28A in VT (**A**–**J**) and EVT (**K**–**O**) in PE with severe features and healthy control group (**P**–**Y**) assessed by immunohistochemistry and double immunofluorescence staining. Red cytoplasmic staining shows LIN28A expression (arrows), while green cytoplasmic staining represents VT and EVT cells positive for CK7. After merging with nuclear DAPI staining, there was no significant difference in expression of LIN28A in EVT and VT in PE compared to that of the control group (arrowhead). Magnification × 40. Scale bar = 25 µm.

**Figure 6 biomedicines-12-00879-f006:**
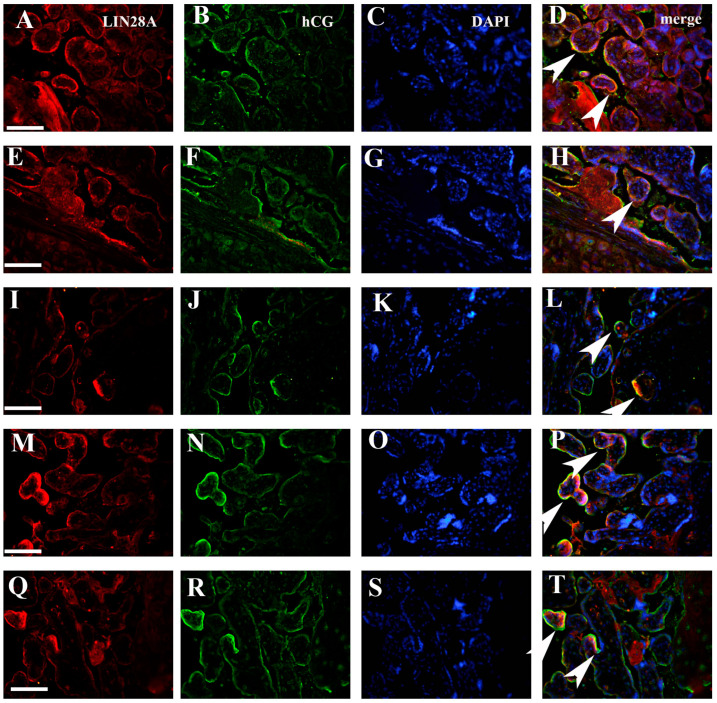
Expression of LIN28A in SCTB in PE with severe features (**A**–**H**) and healthy control group (**I**–**T**) assessed by double immunofluorescence staining. Red cytoplasmic staining shows LIN28A expression, while green cytoplasmic staining shows SCTB cells of chorionic villi positive for hCG. Merging with nuclear DAPI staining revealed no difference in staining intensity between PE and healthy control group (arrowhead). Magnification × 40. Scale bar = 25 µm.

**Figure 7 biomedicines-12-00879-f007:**
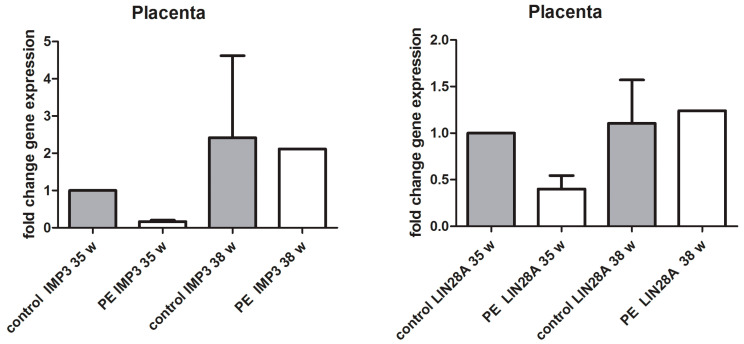
qPCR mRNA fold change of *LIN28A* and *IMP3* from FFPE placental tissue of PE with severe features (35 w) and PE without severe features (38 w) compared to that of healthy controls (35 w and 38 w). Kruskal–Wallis test followed by Dunn’s multiple comparison test was used, but no significant difference was found.

**Table 1 biomedicines-12-00879-t001:** Clinical features of the patients.

	Control 1 (*n* = 10)	PE with Severe Features (*n* = 10)	Control 2 (*n* = 10)	PE without Severe Fetures (*n* = 10)	*p* Value
Maternal age (years) mean ± SD	28.06 ± 5.89	26.50 ± 3.51	27.52 ± 5.19	31.30 ± 5.73	0.390
Gestational age (weeks) median (IqR)	35 (34–38)	34 (33–38)	39 (38–41)	38 (34–40)	<0.001
Systolic RR (mmHg) mean ± SD	115 ± 5.13	178 ± 19.05	119 ± 5.82	148 ± 3.45	<0.001
Diastolic RR (mmHg) median (IqR)	73 (60–85)	123 (110–140)	77 (65–80)	88 (86–100)	<0.001
Birth weight (grams) median (IqR)	2327 (2132–3218)	1818 (1251–2205)	3320 (2350–4515)	3320 (1815–3960)	<0.001
Cesarean deliveries (%)	2 (20)	10 (100)	1 (10)	2 (20)	<0.001
Body mass index (BMI) mean ± SD	24.93 ± 4.78	23.57 ± 3.42	25.64 ± 4.36	23.42 ± 2.36	0.06
Intrauterin growth restriction (IUGR) (%)	1 (10)	7 (70)	0 (0)	1 (10)	<0.001
Postpartal complications (%)	0 (0)	7(70)	0 (0)	1 (10)	<0.001

Data are presented as mean ± SD analyzed by ANOVA, median (IqR = interquartile range) analyzed by Kruskal–Wallis test and percentage using Chi-square test.

**Table 2 biomedicines-12-00879-t002:** Primary antibodies used for immunohistochemistry and double immunofluorescence staining.

Antibody	Dilution	Host	Cellular Localization	Developer
IMP3	1:200	Mouse	Cytoplasm	Dako, Glostrup, Denmark M362629-2
IMP3	1:250	Rabbit	Cytoplasm	Abcam, Cambridge, UK ab179807
LIN28A	1:800	Mouse	Cytoplasm	Cell Signaling Technology Inc, Danvers, MA, USA CT-5930S
LIN28A	1:800	Rabbit	Cytoplasm	Cell Signaling Technology Inc, Danvers, MA, USA CT-3978S
Cytokeratin 7	1:300	Mouse	Cytoplasm	Dako, Glostrup, Denmark M7108
Cytokeratin 7	1.100	Rabbit	Cytoplasm	Ventana Medical Systems Inc., Tucson, AZ, USA SP52
Anti-chorionic gonadotropin	1:50	Rabbit	Cytoplasm	Sigma Aldrich, Sant Louis, MI, USA C8534

**Table 3 biomedicines-12-00879-t003:** Expression of IMP3 and LIN28A in EVT cells of decidua basalis and CTB and SCTB cells of chorionic villi.

Intensity of staining	IMP3		*p*	LIN28A		*p*
	VT			VT		
	PE	CG		PE	CG	
	1.27	2.33	0.002 **	1	1	0.42
Percentage of positive cells	IMP3		*p*	LIN28A		*p*
	EVT			EVT		
	PE	CG		PE	CG	
	26.10	63.35	0.0002 ***	45.2	51.1	0.34

The IMP3 and LIN28A expression is presented as the proportion of positive cells in the total cell count of EVT cells, while the expression in VT cells was presented semi-quantitatively (+1, ++2 or +++3). The chi-square test was used as a statistical method. PE (pre-eclampsia with and without severe features). CG (control group). ** *p* < 0.001; *** *p* < 0.0001.

## Data Availability

The datasets used and/or analyzed during the current study are available from the corresponding author on request.

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
