# Peer review of "Expression of IMP3 and LIN28A RNA-Binding Proteins in Placentas of Patients with Pre-Eclampsia with and without Severe Features"

_biomedicines, 2024, doi:10.3390/biomedicines12040879_

Round 1

Reviewer 1 Report

Comments and Suggestions for Authors

The topic is of high interest, as placenta involvement in different benign pathologies or malignancies pathways is well established. Thus, the study has a great impact upon understanding the placental mechanisms involved in pregnancy disorders, as preeclampsia.

This is an original study which is well presented, well structured, with minor English language errors, especially in Material and Methods section. 

I suggest a minor English language revision for the article.

The results and discussion are well conducted, the microscopic IHC and IF images are of good quality. However, the authors may try to improve the quality of the figures 2 and 7. 

Comments on the Quality of English Language

The quality of English language is good. Only minor corrections are needed.

Author Response

I suggest a minor English language revision for the article.

Thank you for your comment. We amended your advice and revised the English language, focusing on the “Material and Methods” section.

The results and discussion are well conducted, the microscopic IHC and IF images are of good quality. However, the authors may try to improve the quality of the figures 2 and 7. 

Thank you for your comment. We amended your advice and generated new graphs (Figure 2. and Figure 7.) of better quality.

Reviewer 2 Report

Comments and Suggestions for Authors

This is an interesting study. However, some alterations should be made in the terminology. Examples: “Severe preeclampsia” was the former terminology. The current terminology is “Preeclampsia with severe features” (that could include systolic blood pressure ≥160 mmHg and/or diastolic blood pressure ≥110 mmHg on 2 occasions at least 4 hours apart while the patient is on bedrest…).  

Comments on the Quality of English Language

Please wright in a clear and direct style. Example: You can alter the following sentence “The material for analysis was placental tissue, including 10 placentas from women with severe PE, 10 placentas from women with non-severe PE and 20 healthy placentas from term deliveries”, to “The material for analysis was placental tissue, including 10 placentas from women with severe PE, 10 placentas from women with non-severe PE and 20 placentas from healthy women at term” (taking into account the correct terminology). 

Author Response

This is an interesting study. However, some alterations should be made in the terminology. Examples: “Severe preeclampsia” was the former terminology. The current terminology is “Preeclampsia with severe features” (that could include systolic blood pressure ≥160 mmHg and/or diastolic blood pressure ≥110 mmHg on 2 occasions at least 4 hours apart while the patient is on bedrest…).  

Thank you for your comment. We amended your advice, corrected the terminology according to the new ACOG guidelines and accordingly adjusted the reference.

Please wright in a clear and direct style. Example: You can alter the following sentence “The material for analysis was placental tissue, including 10 placentas from women with severe PE, 10 placentas from women with non-severe PE and 20 healthy placentas from term deliveries”, to “The material for analysis was placental tissue, including 10 placentas from women with severe PE, 10 placentas from women with non-severe PE and 20 placentas from healthy women at term” (taking into account the correct terminology). 

Thank you for your comment. We amended your advice.

Reviewer 3 Report

Comments and Suggestions for Authors

Dear Authors,

The authors found a reduced percentage of IMP3 positive EVT cells and low expression of IMP3 in VT of PE placentas of severe and non-severe PE compared with those of healthy controls. The study is relevant to elucidate the pathophysiology of PE.

Before this manuscript is published in journal of “Biomedicines”, some minor corrections should to be made.

1.    In my opinion, the part concerning the expression of RNA binding proteins IMP3 and LIN28A in placental tissue in “Introduction” section should be expanded.

2.     (Table 1) in line 104 should be moved after the first sentence of the paragraph (line 101).

3.     The separation of controls into two groups in Table 1 (Control 1 and Control 2) and the criteria by which they are divided should be described in the text.

4.     The designations (A, B, ... to P) must be clearly visible in Fig. 3.

5.     The “Conclusion” should also include the main result of the study.

Author Response

  1. In my opinion, the part concerning the expression of RNA binding proteins IMP3 and LIN28A in placental tissue in “Introduction” section should be expanded.

Thank you for your comment. We amended your advice and expanded the part regarding the expression of IMP3 and LIN28A in placental tissue in “Introduction” section.

  1. (Table 1) in line 104 should be moved after the first sentence of the paragraph (line 101).

Thank you for your comment. We amended your advice.

  1. The separation of controls into two groups in Table 1 (Control 1 and Control 2) and the criteria by which they are divided should be described in the text.

Thank you for your comment. We amended your advice and added the explanation to the “Materials and Methods” section about how we divided control groups based on the gestational week, to match control groups with the groups of PE with and without severe features. Some previous studies indicated that the expression of investigating proteins may vary depending on the gestational age. Therefore, gestational-age matched control group is required for the appropriate interpretation of the results.

  1. The designations (A, B, ... to P) must be clearly visible in Fig. 3.

Thank you for your comment. We amended your advice and modified figure (Figure 3) accordingly.

  1. The “Conclusion” should also include the main result of the study.

Thank you for your comment. We amended your advice and added the main result of the study to the “Conclusion” section.